# Thin Slabs Made of High-Performance Steel Fibre-Reinforced Cementitious Composite: Mechanical Behaviour, Statistical Analysis and Microstructural Investigation

**DOI:** 10.3390/ma12203297

**Published:** 2019-10-11

**Authors:** Paulo Roberto Ribeiro Soares Junior, Priscila Souza Maciel, Richard Rodrigues Barreto, João Trajano da Silva Neto, Elaine Carballo Siqueira Corrêa, Augusto Cesar da Silva Bezerra

**Affiliations:** 1Department of Materials Engineering, Federal Center for Technological Education of Minas Gerais, Belo Horizonte 30421–169, Brazil; pauloroberto.rsoares@gmail.com (P.R.R.S.J.); richbarrto@gmail.com (R.R.B.); elainecarballo@cefetmg.br (E.C.S.C.); 2Department of Transports Engineering, Federal Center for Technological Education of Minas Gerais, Belo Horizonte 30421–169, Brazil; pmaciel@ymail.com; 3Federal Institute for Education, Science and Technology of Minas Gerais, Ipatinga 35164-261, Brazil; joao.trajano@ifmg.edu.br

**Keywords:** high-performance fibre-reinforced cementitious composite (HPFRCC), flexural strength, toughness, fibre distribution, fibre–matrix interface

## Abstract

The present study evaluated the mechanical behaviour of thin high-performance cementitious composite slabs reinforced with short steel fibres. For this purpose, slabs with 1%, 3% and 5% vol. of steel fibres were moulded using the slurry infiltration method. Fibres concentrated in the region subjected to traction during bending stresses. After curing for 28 days, all slabs underwent flexural testing. The slabs with 5% fibre showed significantly higher flexural strength, deflection and toughness compared to those of the control group without reinforcement. The dense fibre distribution, resulting from the production process, led to profiles with multiple random cracks in the region of failure of the slabs as the fibre content increased. The results of the statistical analysis showed the intensity of the correlation between the variables and revealed that the increase of the fibre content significantly influenced the parameters of mechanical behaviour (load, flexural strength, deflection, toughness and toughness factor). Images obtained by optical microscopy aided in understanding the fibre–matrix interface, showing the bonding surface between the constituents of the composite.

## 1. Introduction

Advances in the areas of infrastructure and construction are closely related to the development of new technologies, especially in terms of advanced materials, durability and new construction processes [1]. In this context, fibre-reinforced cementitious composites (FRCC) emerged at the end of the 19th century in an attempt to create a new cementitious material with two constituents, matrix and reinforcement. The matrix is a paste (cement mixed with water), mortar (paste with sand) or concrete (mortar with large aggregates). The fibres are discrete, randomly oriented and distributed through the composite volume. The matrix and reinforcement work together, producing a synergistic effect and an efficient composite. The addition of fibres allows a desired level of performance regarding a specific property (or properties) [2]. Accordingly, high-performance fibre-reinforced cementitious composites (HPFRCC) have optimized properties and superior performance characterized by improved flexural strength, strain-hardening behaviour led by multiple cracking and relatively large energy absorption [2,3]. Recent studies have been conducted to broaden the knowledge on FRCCs, and many others are under development, given that the subject is broad for new discoveries. 

Mechanical behavior, material characterization, and design considerations of FRCCs have been thoroughly reviewed in comprehensive works [4,5,6]. Several other types of fibres such as sisal [7] and synthetic materials (usually polymers) [8,9] are common alternatives to traditional steel fibres. High fibre contents (5–20%) are present in the so-called slurry-infiltrated fibre concrete (SIFCON) [10,11]. The fibre–matrix interaction is investigated to understand the behaviour of composite phases through pull-out tests and evaluation of associated phenomena such as adhesion, friction and anchoring [12,13,14,15]. The influence of additions is pertinent, as it changes the interfacial transition zone (ITZ), contributing to fibre bonding in the matrix [16,17]. The use of hybrid fibres has been promising because it combines different fibre properties, generating an optimized synergistic effect [18,19,20]. The distribution and orientation of fibres in the matrix has been shown to be an influential factor in composite behaviour, so it has been the focus of recent studies [21,22]. Impact loads imply high loading rates and alter a material response. In this case, FRCCs are an effective alternative to combat cracking, which promotes performance increases [23]. Fire resistance and very low temperatures can be increased by the addition of fibres. In the case of high temperatures with high heating rates, the fibres restrict spalling [24,25,26]. Hydraulic shrinkage occurs during curing of cementitious materials and can be reduced by the presence of fibres [27]. Fibrous reinforcement can also be used in self-compacting concretes, expanding the possibilities of use of the composite [28]. Prediction models have been used to aggregate information and experimental data [29]. In addition to the search field presented, another aspect has emerged to combine sustainability concepts with FRCCs. In this regard, attempts are made to make the composite eco-friendly, using natural aggregates, recycled fibres, cements and sustainable additions [30,31]. 

Throughout the development of HPFRCCs, SIFCON emerged. SIFCON is characterized by a high fibre content, a particular moulding process and high capacity to absorb energy [10,11,32]. The production of SIFCON consists of inserting the reinforcement fibres into the moulds and subsequently pouring a fluid mortar to fill the voids between the fibres [33]. The result is a composite with a dense fiber distribution and excellent mechanical behaviour [34]. The use of SIFCON is widespread and can be applied to repairs and reinforcements, pavements, precast elements and structures resistant to seismic tremors [35,36].

Several studies have addressed the flexural strength of FRCCs, considering the different characteristics of the matrix, the reinforcement and the interfacial transition zone [32,37,38,39,40]. According to Naaman and Reinhardt [41], the mechanical behaviour under bending can be classified as deflection hardening when the composite withstands a greater load after the point of first fracture or deflection softening when it presents a strength lower than the load withstood by the matrix at the time of cracking. Generally, the tests are performed in a reduced model with some simplifications, using prismatic specimens and square sections. Other models are also adopted, such as panels and slabs, but they are less common [4].

The mechanical parameters during bending (i.e., flexural strength, deflection capacity and toughness) provide the characteristics of the studied material and are influenced by several factors: curing conditions, presence of large aggregates and additions, fibre length and content, geometry and loading rate during the test [4]. The effect of size and the fibre distribution are other sensitive parameters. Reduction in the size of the specimens accentuates the fibre alignment tendency during the moulding process, which favours mechanical performance [42]. The distribution and orientation of the fibres are influenced by factors such as matrix fluidity, pouring method and volumetric fraction [43]. Eventually, the distribution of fibres may be intentional, which results in specific and individual behaviours [31].

In this scenario, it is evident that HPFRCCs have several applications and a great potential for new discoveries. Several studies have addressed the mechanical behaviour of HPFRCCs in general using conventional specimens that represent more robust structures with fibre distribution throughout the composite volume. However, experimental approaches with narrow models combined with the use of concentrated fibres are still sparse in recent studies. Thus, the present study evaluated the mechanical behaviour of thin slabs made of a high-performance steel fibre-reinforced cementitious composite. Focusing on the flexural mechanical behaviour, a new model with slender sections and fibres concentrated in the region where tensile stresses are applied was evaluated.

## 2. Materials and Methods

The materials used were Portland cement of high early strength (type III according to ASTM C150/C150M [44]), active silica, washed natural sand, superplasticizer, short steel fibres and water. The methodology of preparation and analysis of the specimens included dosing of fluid mortar, moulding, flexural strength tests, statistical analysis and fibre–matrix analysis. Steel fibres with anchored ends were used for reinforcement of the cementitious matrix. The properties of the fibres are shown in Table 1.

River sand was used as the fine aggregate, which had a maximum grain size of 4.8 mm, promoted strength gain and contributed to the increase in matrix density. A polycarboxylate-based superplasticizer modified with stabilized nanosilica was used to increase the fluidity of the mortar and allow the flow of material through the dense fibre distribution in SIFCON [45].

Beglarigale et al. [46] used an optimal proportion of cementitious materials, mineral additives, large aggregates and water. By following the same concept and experimentally adjusting the dosage of the constituents, with the inclusion of a superplasticizer, the constant ratio shown in Table 2 was obtained.

The specimens had rectangular faces, slender sections and dimensions of 330 × 165 × 25 mm^3^ (length, width and thickness). Moulding was performed following the slurry infiltration method [11,32]. The fibres were placed in the moulds at 1% vol., 3% vol. and 5% vol. The fluid mortar was then poured to fill the spaces between the fibres (Figure 1). Two specimens were prepared for each fibre content.

After 28 days of curing in a solution saturated with calcium hydroxide (1.7 g per litre of water), the specimens were subjected to a four-point flexural strength test. The tests were performed in an electronically controlled universal testing machine (EMIC DL30000, Belo Horizonte, Brazil), with a loading rate equal to 0.1 MPa/s and a distance between supporting pins (L) equal to 240 mm. The values of applied load and deflection at the midpoint of the span were measured and adjusted in the form of loading curves. The stress values were calculated as established in the standard [47]. The characteristic load, stress and deflection values were obtained for the point of first cracking (P_LOP_, σ_LOP_ and δ_LOP_), which is called limit of proportionality (LOP), and for the maximum loading (P_u_, MOR and δ_MOR_), which is identified as the modulus of rupture parameters. The toughness (T) was calculated as the area under the load-versus-deflection curve for the characteristic deflections of L/150 and L/40 and for the maximum deflection. The toughness factor was calculated according to Equation (1) [37,48,49]:(1)$$FT=\frac{T\cdot L}{\frac{L}{150}\cdot b\cdot h^{2}}$$
where b is the width of the specimen, and h is its height. The toughness factor FT in J/m³ was then obtained and is the average withstood load after matrix fissuring.

A statistical analysis was performed to evaluate the significance of the data obtained and the correlation among the response variables (load, stress, deflection, toughness and toughness factor). First, the correlation among the variables was measured by the correlation coefficient. Multivariate analysis of variances (MANOVA) was used to evaluate the influence of fibre addition, considering the response variables together. Finally, univariate analysis of variance (ANOVA) was used to verify the significance of the data in terms of the response variables individually, i.e., ANOVA does not consider the correlation among the response variables. The magnitude of the effect represents the impact of the categorical variable (fibre content) on the response variables; its value is greater than zero, and values of approximately 1 or higher indicate a broad effect. The power of the test shows the robustness of the tests in identifying differences between the evaluated groups; it ranges from 0 to 1, and values above 0.8 are considered reasonable [50,51]. The significance level adopted is 5% (0.05); therefore, the analyses have a confidence interval equal to 95%. A fibre–matrix analysis was performed using optical microscopy to evaluate the composite elements (matrix and reinforcement) and the interfacial transition zone.

## 3. Results and Discussion

### 3.1. Flexural Strength

Figure 2 shows a characteristic load-versus-deflection curve for each fibre content and the average maximum strength. The consolidated results for all curves are shown in Table 3. The characteristic deflection values for the calculation of toughness were identified (L/150 and L/40). The fibre-reinforced composites showed a “deflection-hardening” behaviour after the LOP, with strength gain along the deflections, even with the cracked matrix. The “deflection-softening” behaviour was observed after reaching the maximum load (modulus of rupture, MOR). In general, when the fibres were arranged in the tensile stress region, the fibre content significantly influenced the flexural strength parameters. The LOP increased from 3.46 MPa (matrix without reinforcement) to 19.04 MPa (5% fibre content) with the incorporation of reinforcement, corresponding to a 5.5–fold increase. The MOR, depicted by the mean flexural strength (Figure 1b), increased with increasing fibre content. Table 3 shows that by increasing the addition of fibres from 1% to 5%, there was a 3.5–fold increase in the modulus of rupture.

Flexural strength analysis showed that the addition of fibres was directly correlated to the improvement of the mechanical behaviour of the material. Table 3 shows that the mean value for the control group was 3.46 MPa. For the models with 1%, 3% and 5% fibres, the mean flexural strengths were 5.84 MPa, 16.33 MPa and 21.02 MPa, respectively. This is a six-fold increase in the flexural strength for the model with 5% fibres compared to the model without fibres. This is because the fibres distributed in the mortar withstood the formation of cracks and prevented their growth. After the first crack, the models with fibre addition exhibited post-cracking strength because the fibres “stitched” the cracks by transferring the loads. The differentiated arrangement in the lower area of the model was crucial for this behaviour, because the fibres acted on the most required mechanical region.

The model without fibres ruptured suddenly at small deflection values compared to the other models, thus lacking a toughness factor. This characteristic was due to the fragile nature of the mortar, which prevented it from withstanding considerable deflections; the propagation of cracks due to the accumulation of stresses in the fractures was the rupture factor. For the models with the addition of fibres, there was an increase in the absorption energy until rupture due to their ability to withstand deflections and loads, therefore increasing toughness. For the addition of 1% fibre, the mean toughness was 1.28 J for L/150 and 3.55 J for the maximum deflection. For the addition of 3% fibre, it was possible to calculate the toughness for L/150 (5.05 J), L/40 (18.87 J) and the maximum deflection (21.04 J). Therefore, by tripling the amount of fibres, there was a six-fold increase in toughness for the maximum deflection. The increase in toughness for the model with 5% fibres was more than twice than that for the 3% fibre model.

The toughness factor was obtained from the toughness, specimen dimensions, distance between supporting pins and deflection withstood. The toughness factor is related to the post-cracking load, where the plastic deformation regime and the fibre–matrix interaction mechanism prevail [52,53,54]. The values for the model without fibres were much lower than those for the model with the addition of fibre. As expected, the higher the fibre addition was, the higher the toughness factor obtained.

The deflections increased considerably with increasing fibre content, which revealed the pseudo-ductile behaviour of the composite. Compared to the control group, the addition of 5% fibres resulted in deflections up to 12 times higher.

Figure 3 shows the cracking profile of the slabs after the bending test. At first glance, the cracks extend along the width of the slabs in a direction perpendicular to the soliciting tensile stresses. The slabs without reinforcement (Figure 3a) ruptured abruptly, showing the unstable propagation of cracks that is characteristic of cementitious materials [55,56]. In this case, a single narrow crack was found, which extended along the slab. In contrast, the slabs reinforced with fibres (Figure 3b–d) experienced progressive failure. A main crack was observed along the slab, which governed the failure mechanism, followed by multiple adjacent cracks. After the first cracking stage, the interaction mechanism between the fibre and the matrix prevailed, and strains occurred gradually until failure of the slab [4,57]. It was observed that the higher the fibre content, the less noticeable the main crack. The adjacent cracks grew in number and became more random, indicating the transition from a material with fragility characteristics to an increasingly pseudo-ductile composite [2,58,59]. The release of cementitious material, especially in the composite with higher fibre contents (Figure 3d), can be explained by the cracking of the matrix and the relative displacement of the fibres during deflection. The mechanisms of cracking and the presence of random cracks are closely related to the dense fibre distribution existing in SIFCON, which promotes a robust, cohesive composite with a great capacity for energy absorption [11,32,33,34,35,36]. In this context, it is possible to correlate the failure mode with the mechanical behaviour evaluated by the load-versus-deflection curves [2,4,58,59]. The highest values of strength, deflection and toughness were associated with slabs with extensive cracking and high fibre content (Figure 3d), where the mechanism of fibre–matrix interaction prevailed. In contrast, a well-defined single crack and an abrupt failure of the slab corresponded to the lowest values of the mechanical parameters.

### 3.2. Statistical Analysis

Table 4 shows the correlation coefficients among the mechanical parameters. The represented values form a square matrix symmetric in relation to the main diagonal. Each row or column shows the correlation of a given variable with the others. The values of the main diagonal represent the correlation between a variable and itself. As expected, all these values are equal to 1 because, in this case, the correlation is perfect. When comparing two parameters, the proximity of the correlation coefficient to the unit indicates the intensity of correlation between the parameters. In general, the parameters are strongly correlated, which justifies an approach by means of MANOVA. The deflections, both δ_LOP_ and δ_MOR_, had coefficients ranging from 0.643 to 0.852, while the other parameters had values above 0.9. All values were statistically significant; however, the parameters load, stress, toughness and toughness factor were more influential on each other than the deflections. It is known that using only two specimens, as done here, is statistically feasible but decreases the robustness of the model. However, the statistical analysis is relevant mainly because it allows an analysis between the response variables.

Table 5 shows the results of the MANOVA. The results are significant for all three tests used (*p* < 0.05). The effect size was large and showed that the fibre content had a significant impact on the correlated response variables. The power of the test was consistent and showed that the tests used were effective in identifying the differences between the groups.

Table 6 shows the ANOVA results. Considering the analysis of the parameters individually, the tests were significant (*p* > 0.05), i.e., the variation of the fibre content significantly influenced the mechanical behaviour. The effect size was significant and similar for all parameters, and the power of the test was consistent.

### 3.3. Fiber–Matrix Analysis

Figure 4 shows optical microscopy images, with a focus on the fibre–matrix analysis. The anchored end of the fibre is observed in Figure 3a. Despite the circular shape of the fibres, there are flattened points due to the manufacturing and conformation processes of the anchored ends [60,61]. The anchoring of the fibres helped with the fixation process, promoting the anchoring of the fibre in the matrix. Increased adhesion generates greater strength regarding crack propagation. The interaction capacity between the matrix and the reinforcement is linked to the relationship between the fibre length and the maximum aggregate diameter as well as the fibre aspect ratio [4,58,59]. Small pores were observed in the matrix due to the loss of free water and the incorporation of air in the mixing process [62]. Figure 3b shows the straight section of a fibre enveloped in the cementitious matrix. The fibre surface has a corrugated appearance due to the pull-out mechanism and interaction with the matrix, associated with the friction mechanisms and plastic deformations of the fibre. When the fibre is fully bonded, the rectilinear segment is responsible for contact adhesion, and the curved segment (Figure 4a) anchors in the matrix. During the pull-out process, after complete disconnection, the rectilinear segment of the fibre slides into the formed duct, and the bonding forces are attributed to the friction. In the hook section, the behavior is more complex. The curved segments are forced through the ducts and bend successively. Therefore, at this stage, the pull-out process is governed by plastic deformations of the fibre. [63]. In the matrix, it is possible to see fragments of crystalline quartz due to the quartz nature of the sand used as a small aggregate [64]. Figure 3c shows fibres distributed randomly, in addition to mortar fragments detached from the matrix due to material failure caused by excessive strains. Ducts left by the fibres in the matrix are observed, and the remnants of residual mortar show adhesion between fibre and matrix [65]. Figure 3d shows the niche of the fibre–matrix bezel and several micro-cracks that extend throughout the matrix. The proximity between the fibres reveals the dense discrete reinforcement distribution present in SIFCON [32,34] (Figure 3c,d). The fibre–matrix bonding niche is closely related to the non-oriented fibre distribution, especially when the infiltrated grout method is used. Very close fibres are evident in the bonding niche, which generates a synergistic effect and influences the behavior of the composite. 

## 4. Conclusions

The present study investigated thin slabs made of HPFRCC, focusing on their mechanical behaviour. The following conclusions can be drawn from the experimental programme, results and discussions:(1)The flexural strength of thin slabs made of HPFRCC was evaluated in terms of load-versus-deflection curves. The mechanical parameters were calculated and evaluated (load, stress, deflection, toughness and toughness factor). In general, the fibre content affected all parameters. The higher the fibre content, the higher the values obtained for the parameters of mechanical behaviour. The toughness reached values 120 times higher than those of the control group, and the toughness factor was approximately 12 times higher than that of the control. The load-versus-deflection curves showed the deflection hardening mechanism, considering that the modulus of rupture was much greater than the limit of proportionality. Thus, after the point of first cracking, the composite continued to gain mechanical strength until it reached the limit of strength. This performance resulted from the mechanisms of fibre–matrix interaction and progressive deformations with multiple cracks.(2)In general, the cracking profile extended along the width of the slab and reached the entire cross section. For the slabs without reinforcement, only a single crack was observed, showing fragile behaviour. For slabs with reinforcement, a main crack extended over the width of the slab and governed the failure mechanism, and multiple adjacent cracks with random orientation appeared. This mechanism may be related to mechanical parameters and load-versus-deflection curves. The greater the fibre content is, the greater the strength, the deflections and the toughness and the more numerous the adjacent cracks are, indicating the transition from fragile to pseudo-ductile behaviour.(3)Statistical analyses showed that the mechanical parameters were clearly influenced by fibre content and that this relationship was significant for both MANOVA and univariate ANOVA. Furthermore, it was possible to verify that the mechanical parameters were strongly correlated, i.e., the variation of one of them implies the variation of the others. The power of the tests was expressive; thus, the analysis model adopted satisfactorily explains the evaluated problem.(4)The fibre-matrix analysis revealed the constituents of the composite (matrix, fibres and interface). The anchored ends of the fibres helped with the adhesion process. The presence of pores in the matrix, resulting from the release of free water and incorporation of air, and the presence of quartz crystals due to the use of quartz sand were observed. The ducts left by the fibres in the matrix became apparent after mechanical testing, and loose fragments of cementitious matrix and several microcracks were observed in the continuous phase of the composite. The presence of residual mortar adherent to the fibres indicated adhesion between the fibre and the matrix.(5)Further research could clarify the fibre–matrix binding mechanism, and the pull-out assay may be suggested for further investigation. The influence of the dense fibre distribution should also be investigated.

## Figures and Tables

**Figure 1 materials-12-03297-f001:**
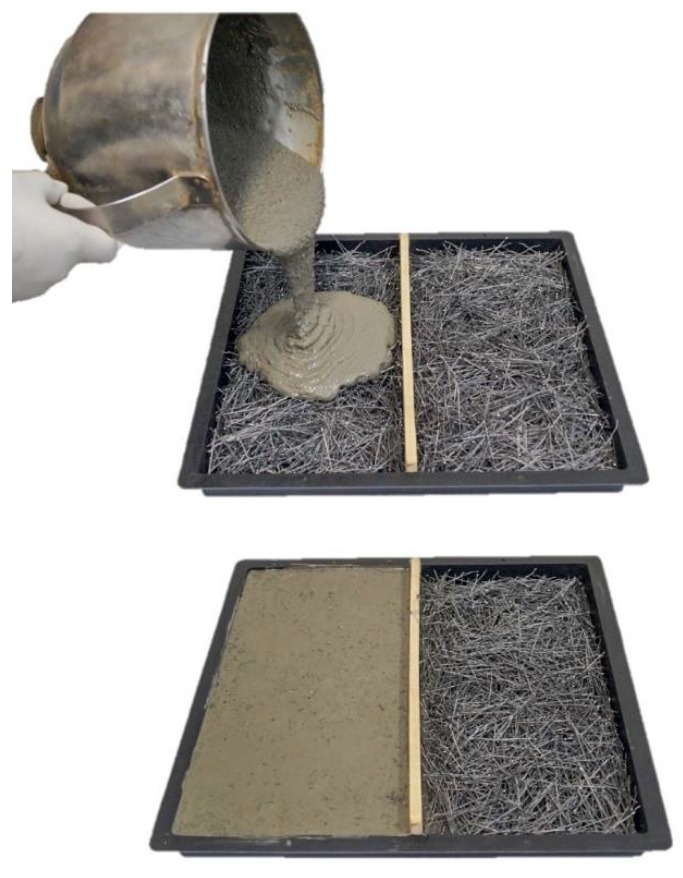
Slurry infiltration method.

**Figure 2 materials-12-03297-f002:**
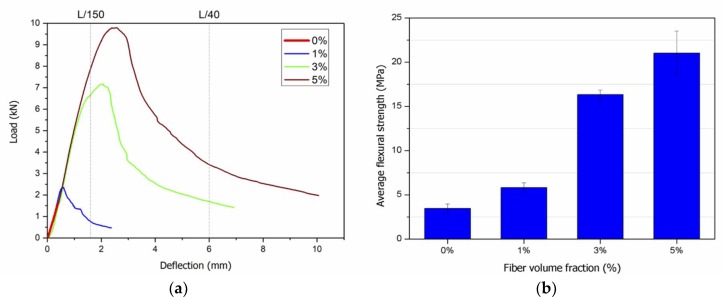
Characteristic load-versus-deflection curves and (**b**) average flexural strength.

**Figure 3 materials-12-03297-f003:**
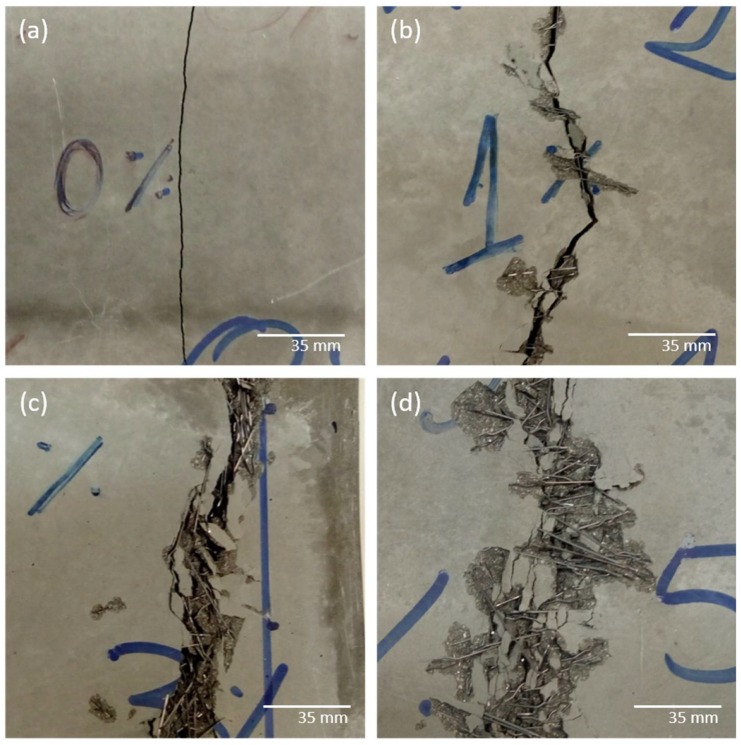
Crack profile of the slabs (**a**) without reinforcement, (**b**) 1% fibre, (**c**) 3% fibre and (**d**) 5% fibre.

**Figure 4 materials-12-03297-f004:**
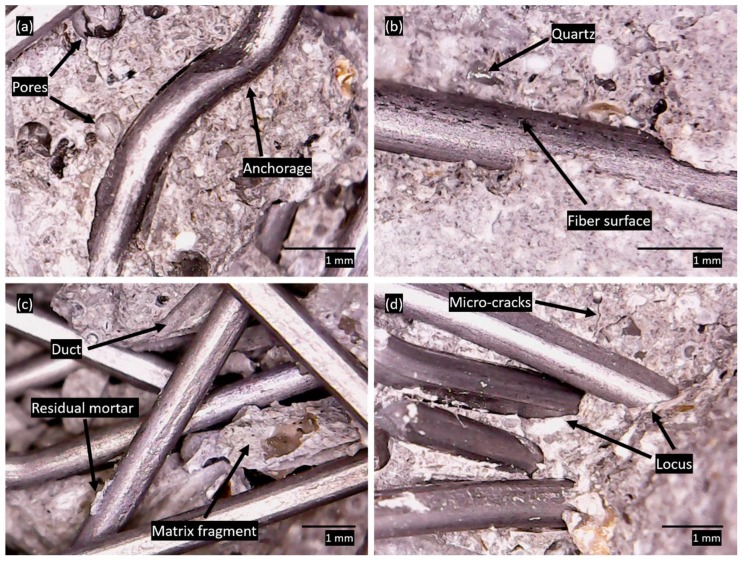
Images obtained by optical microscopy after mechanical testing: (**a**) anchored fibre segment, (**b**) fibre–matrix adhesion in the straight section, (**c**) non-oriented distribution of the fibre in the matrix and (**d**) fibre–matrix bond niche.

**Table 1 materials-12-03297-t001:** Characteristics of steel fibres.

Characteristic	Steel Fibre
Material	Low-carbon steel
Manufacturing method	Cold forming (wire drawing)
Section shape	Circular
Length (mm)	50.35
Diameter (mm)	0.72
l/d (shape factor)	68.79
Tensile strength	1350 MPa

**Table 2 materials-12-03297-t002:** Proportion of mortar constituents.

Materials Proportion (kg/m^3^)				Factor a/c
Binder	Aggregate	Silica Fume	Superplasticizer	
831.5	831.5	166.3	12.5	0.4

**Table 3 materials-12-03297-t003:** Parameters of the mechanical behaviour under bending.

F (%)	SP	Mechanical Behaviour Parameters												
		P_LOP_	σ_LOP_	δ_LOP_	P_u_	MOR	δ_MOR_	δ_MAX._	T_δmax_	T_L/150_	T_L/40_	FT_δmax_	FT_L/150_	FT_L/40_
		(kN)	(MPa)	(mm)	(kN)	(MPa)	(mm)	(mm)	(J)	(J)	(J)	(J/m^3^)	(J/m^3^)	(J/m^3^)
0%	SP1	1.64	3.82	0.84	1.64	–	–	0.84	0.34	–	–	0.94	–	–
	SP2	1.33	3.09	1.03	1.32	–	–	1.03	0.20	–	–	0.46	–	–
	Mean.	1.49	3.46	0.94	1.48	–	–	0.94	0.27	–	–	0.7	–	–
1%	SP1	2.28	5.31	0.52	2.35	5.47	0.62	2.96	2.63	1.53	–	2.07	2.22	–
	SP2	2.16	5.03	0.42	2.67	6.21	0.98	4.35	4.47	1.03	–	2.39	1.51	–
	Mean.	2.22	5.17	0.47	2.51	5.84	0.80	3.66	3.55	1.28	–	2.23	1.87	–
3%	SP1	5.83	13.57	1.22	7.18	16.71	2.05	6.91	22.51	5.77	19.48	7.58	8.40	7.55
	SP2	5.49	12.79	1.49	6.85	15.95	2.04	5.73	19.57	4.33	18.26	7.95	6.30	7.08
	Mean.	5.66	13.18	1.36	7.02	16.33	2.05	6.32	21.04	5.05	18.87	7.77	7.35	7.32
5%	SP1	8.60	20.02	1.68	9.79	22.79	2.55	10.05	45.25	6.16	30.42	10.48	8.97	11.80
	SP2	7.76	18.05	1.49	8.27	19.25	1.70	10.62	41.89	7.00	30.57	9.18	10.18	11.86
	Mean.	8.18	19.04	1.59	9.03	21.02	2.13	10.34	43.57	6.58	30.50	9.83	9.58	11.83

Legend: F—fiber content; SP—specimen; LOP—limit of proportionality; MOR—modulus of rupture; P_LOP_—load at the LOP; σ_LOP_—stress at the LOP; δ_LOP_—deflection at the LOP; P_u_—maximum load; _MOR_δ—deflection at the MOR; δ_MAX_—maximum deflection; T_δmax_—toughness for maximum deflection; T _L/150_—toughness for deflection equal to L/150; T _L/40_—toughness for deflection equal to L/40; FT_δmax_—toughness factor for maximum deflection; FT_L/150_—toughness factor for deflection equal to L/150; FT_L/40_—toughness factor for deflection equal to L/40.

**Table 4 materials-12-03297-t004:** Coefficient of correlation between the mechanical parameters.

Correlation Coefficients													
	P_LOP_	σ_LOP_	δ_LOP_	P_u_	MOR	δ_MOR_	δ_MAX._	T_δmax_	T_L/150_	T_L/40_	FT_δmax_	FT_L/150_	FT_L/40_
**P_LOP_**	1	1.000	0.743	0.976	0.976	0.714	0.929	0.976	0.976	0.976	0.952	0.976	0.976
**σ_LOP_**	1.000	1	0.743	0.976	0.976	0.714	0.929	0.976	0.976	0.976	0.952	0.976	0.976
**δ_LOP_**	0.743	0.743	1	0.719	0.719	0.850	0.707	0.719	0.707	0.707	0.755	0.707	0.707
**P_u_**	0.976	0.976	0.719	1	1.000	0.762	0.952	1.000	0.952	0.952	0.976	0.952	0.952
**MOR**	0.976	0.976	0.719	1.000	1	0.762	0.952	1.000	0.952	0.952	0.976	0.952	0.952
**δ_MOR_**	0.714	0.714	0.852	0.762	0.762	1	0.738	0.762	0.643	0.643	0.738	0.643	0.643
**δ_MAX._**	0.929	0.929	0.707	0.952	0.952	0.738	1	0.952	0.952	0.952	0.929	0.952	0.952
**T_δmáx_**	0.976	0.976	0.719	1.000	1.000	0.762	0.952	1	0.952	0.952	0.976	0.952	0.952
**T_L/150_**	0.976	0.976	0.707	0.952	0.952	0.643	0.952	0.952	1	1.000	0.929	1.000	1.000
**T_L/40_**	0.976	0.976	0.707	0.952	0.952	0.643	0.952	0.952	1.000	1	0.929	1.000	1.000
**FT_δmax_**	0.952	0.952	0.755	0.976	0.976	0.738	0.929	0.976	0.929	0.929	1	0.929	0.929
**FT_L/150_**	0.976	0.976	0.707	0.952	0.952	0.643	0.952	0.952	1.000	1.000	0.929	1	1.000
**FT_L/40_**	0.976	0.976	0.707	0.952	0.952	0.643	0.952	0.952	1.000	1.000	0.929	1.000	1

**Table 5 materials-12-03297-t005:** Results of multivariate analysis of variance (MANOVA).

MANOVA			
Test	Sig.	Size Effect	Power
Pillai’s Trace	0.001	0.923	0.997
Wilks’ Lambda	0.029	0.971	0.609
Roy’s Largest Root	0.001	0.996	1.000

**Table 6 materials-12-03297-t006:** Results of the univariate analysis of variance (ANOVA).

Parameter	Sig.	Effect Size	Power
P_LOP_	0.000	0.992	1.000
σ_LOP_	0.000	0.992	1.000
δ_LOP_	0.005	0.949	0.991
P_u_	0.001	0.983	1.000
MOR	0.001	0.983	1.000
δ_MOR_	0.030	0.871	0.753
δ_MAX._	0.001	0.981	1.000
T_δmáx_	0.000	0.995	1.000
T_L/150_	0.001	0.973	1.000
T_L/40_	0.000	0.999	1.000
FT_δmáx_	0.000	0.991	1.000
FT_L/150_	0.002	0.971	1.000
FT_L/40_	0.000	0.997	1.000

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
