# Peer review of "Thin Slabs Made of High-Performance Steel Fibre-Reinforced Cementitious Composite: Mechanical Behaviour, Statistical Analysis and Microstructural Investigation"

_materials, 2019, doi:10.3390/ma12203297_

Round 1

Reviewer 1 Report

General comment: please try to rephrase some of the paragraphs and have a general English check of the text.

Please fin below a list of suggestions on how to improve the paper:

Line 42: “superior performance” – in terms of?

Line 43: studies have been conducted not developed

Line 43: reference 4-29 – please enunciate the broaden knowledge. The authors chose to include 25 references in the list but did not mention the contribution of the research works (in smaller groups if possible) to the state of the art.

Line 79 – superplaticizer is enough

Line 80 – methods include? Please rephrase as it is rather unclear

Line 93 – a photo showing the casting process would be much more relevant

Line 104 – proportionality limit

Figure of the loading layout and the location of the measuring points

Equation 1 – T, the toughness should be computed from the load-deflection curve not stress-deflection curve [48]. Otherwise, the units of measure do not match. L/150 is a non dimensional factor and this should be mentioned when explaining the terms in the equation. Again, this could lead to a mismatch of the units of measure

Line 139, Line 150 – Table 3 instead of Table 4

Table 3 – one of the major concerns is the reduced number of specimens / investigate case. Two specimens is not enough to perform a sound statistical analysis. The norms recommend at least 3 values (for conformity checks) so that the extremes are discarded and the statistical analysis is performed on the remaining data. Concrete, in all its forms and mixes, is a heterogeneous material and may lead to large scattering of the results. The fact that the authors obtained good values of some correlation coefficients is, in my opinion, by pure luck. The number of specimens should be reconsidered in order for a sound statistical analysis to be performed.

Line 157 – most loaded part of the cross-section in tension; “solicited” is not often use to describe this case.

Line 160 – strain or deflection (Line 165)? Concrete surface strain gages should have been used as the calculations of strains, in this case, is rather cumbersome and may lead to erroneous results.

Line 171 – withstood? Toughness factor is also related to the post-peak / fracture behavior of the specimen. There is no such behavior for the specimens without fibers (Line173)

Reviewer 2 Report

I appreciate the Editor to give me a chance to review an interesting and valuable paper. I found some merits in the both methodology and results. In my opinion, this paper has a good potential to be published in the journal. However, I have also some concerns on the different parts of the manuscript. If the author(s) address carefully to all of my comments, I’ll recommend publication of the manuscript in the journal:

·        Discuss the differences between anchored fiber segment, fiber-matrix adhesion in the straight section, non-oriented distribution of the fiber in the matrix and fiber-matrix bond niche.

·        It is necessary to explain the sources of error in this study to consider them in next investigations.

·        At the end of the manuscript, explain the implications and future works considering the outputs of current study.

·        The quality of the language needs to improve by a native English speaker for grammatically style and word use.

Round 2

Reviewer 1 Report

The manuscript has been improved, especially from the point of view of added explanations and reworked figures. The message is much more clear now and the possible misunderstandings have been significantly reduced.

The authors are strongly encouraged to continue in this line of research and move past the state of the art. The topic is of large interest in the era of sustainable development.

Reviewer 2 Report

I appreciate the authors for addressing the comments. The quality of the manuscript is acceptable now.